# Elucidation of Pharmacological Mechanism Underlying the Anti-Alzheimer’s Disease Effects of *Evodia rutaecarpa* and Discovery of Novel Lead Molecules: An In Silico Study

**DOI:** 10.3390/molecules28155846

**Published:** 2023-08-03

**Authors:** Lulu Zhang, Jia Xu, Jiejie Guo, Yun Wang, Qinwen Wang

**Affiliations:** 1State Key Laboratory of Medical Neurobiology, MOE Frontiers Center for Brain Science, Institutes of Brain Science, Department of Neurology, Zhongshan Hospital, Fudan University, Shanghai 200032, China; 19111520017@fudan.edu.cn; 2Ningbo Key Laboratory of Behavioral Neuroscience, Zhejiang Provincial Key Laboratory of Pathophysiology, School of Medicine, Ningbo University, Ningbo 315211, China; xujia@nbu.edu.cn (J.X.); gjj_19880608@126.com (J.G.)

**Keywords:** *Evodia rutaecarpa*, pharmacology network, Alzheimer’s disease, migraines, neurotransmitter inflammation, hormones

## Abstract

Alzheimer’s disease (AD) is a brain disease with a peculiarity of multiformity and an insidious onset. Multiple-target drugs, especially Chinese traditional medicine, have achieved a measure of success in AD treatment. *Evodia rutaecarpa* (Juss.) Benth. (Wuzhuyu, WZY, i.e., *E. rutaecarpa*), a traditional Chinese herb, has been identified as an effective drug to cure migraines. To our surprise, our in silico study showed that rather than migraines, Alzheimer’s disease was the primary disease to which the *E. rutaecarpa* active compounds were targeted. Correspondingly, a behavioral experiment showed that *E. rutaecarpa* extract could improve impairments in learning and memory in AD model mice. However, the mechanism underlying the way that *E. rutaecarpa* compounds target AD is still not clear. For this purpose, we employed methods of pharmacology networking and molecular docking to explore this mechanism. We found that *E. rutaecarpa* showed significant AD-targeting characteristics, and alkaloids of *E. rutaecarpa* played the main role in binding to the key nodes of AD. Our research detected that *E. rutaecarpa* affects the pathologic development of AD through the serotonergic synapse signaling pathway (SLC6A4), hormones (PTGS2, ESR1, AR), anti-neuroinflammation (SRC, TNF, NOS3), transcription regulation (NR3C1), and molecular chaperones (HSP90AA1), especially in the key nodes of PTGS2, AR, SLCA64, and SRC. Graveoline, 5-methoxy-N, N-dimethyltryptamine, dehydroevodiamine, and goshuyuamide II in *E. rutaecarpa* show stronger binding affinities to these key proteins than currently known preclinical and clinical drugs, showing a great potential to be developed as lead molecules for treating AD.

## 1. Introduction

Alzheimer’s disease (AD) is a worldwide disease, and its etiology is still not precisely known [1,2]. The pathological mechanism of AD shows multiformity and an insidious onset [3], mainly related to the amyloid-β peptide (Aβ) [4], the tau protein [5], intraneuronal neurofibrillary tangles (NTFs), lipid metabolism [6,7], energy metabolism [8], and neuroinflammation [9,10]. In addition, neurotransmitter disorders [10], hormone imbalances [11], and genetic components [12] are also considered to play important roles in the occurrence and development of AD. Over the last decade, many clinical trials, including several phase III trials, have been conducted to find safe and effective drugs for AD, but the results have not met expectations [3,11,12,13,14,15].

Natural herbs have been widely used in traditional Chinese medicine (TCM) for thousands of years. Recently, TCM has received widespread attention in the world, particularly given its few side effects, multi-target treatments, and natural origin [16,17]. A growing number of studies have indicated that some kinds of natural herbs with multi-target characteristics can successfully treat AD in both in vivo and in vitro models [18,19]. Natural herbs have shown hope and great potential for the treatment of AD [20,21]. The dried and nearly ripe fruit of *E. rutaecarpa* (Juss.) Benth. (i.e., *E. rutaecarpa*) is a natural herb with many therapeutic values: it has analgesic, anti-inflammatory, anti-anxiety, anti-depression, and lipid-metabolism-regulating properties and has been used to treat migraines, painful menstruation, abdominal pain, emesis, diarrhea, and cardiovascular diseases in TCM clinics. *E. rutaecarpa* is rich in alkaloids, containing mainly indole alkaloids, quinolone alkaloids, limonin, and flavonoids [16,22,23,24]. The alkaloids contained in *E. rutaecarpa* have been reported to have anti-Alzheimer’s disease effects. Dehydroevodiamine can attenuate amnesia induced by Aβ (25–35) or scopolamine in mice, ameliorate spatial memory retention impairment in rats [17], and decrease tau hyperphosphorylation [18]. Evodiamine, a main compound in *E. rutaecarpa*, has neural protective effects: it improves cognitive ability [19], decreases the neurotoxicity of tau aggregation [25], inhibits glial cell activation and neuroinflammation [26], suppresses oxidative stress, and increases serum levels of acetylcholine choline acetyltransferase [27]. Moreover, berberine can accelerate cell viability and suppress caspase-3 activity and apoptotic rate [28] and protect against cognitive deficits by improving tau hyperphosphorylation and axonal damage [29]. In addition, there are also many other natural active compounds contained in *E. rutaecarpa* whose medicinal values and potential in the treatment of AD are not yet known.

A recent study indicated that *E. rutaecarpa* extract could effectively improve the behavioral performance of learning and memory in AD model mice [17]. There is no clear record of *E. rutaecarpa* treating AD, though it has been proven to treat migraines and depression in TCM. Correspondingly, in our initial study, through QSAR target prediction and Disease Ontology (DO) enrichment analyses, we discovered that *E. rutaecarpa* predominantly targeted AD-related targets, rather than migraines and depression. In addition, in our previous study, we found that a kind of decoration (patent pending) contained *E. rutaecarpa* as a main effective component, which could be helpful for decreasing the damage to learning and memory abilities caused by Aβ in rats. Based on the above research, we hypothesize that *E. rutaecarpa* may have a medicinal value for AD treatment, which may be related to some of the effective compounds contained within it.

In silico techniques have made many achievements in the screening of lead molecules and drug discovery [18,19,25,26,27,28]. In this work, we employed in silico techniques, including compound collection, target prediction, enrichment analyses, pharmacology network construction, hub node identification, and molecular docking, to confirm and elucidate the mechanism underlying the anti-AD effects of *E. rutaecarpa*. In the collection of compounds, we collected compounds from two main databases for comprehensiveness. Meanwhile, standard ADME screening and QSAR target prediction were utilized for accurate and comprehensive collection and prediction. Then, we overlapped these predicted targets with an AD-related target collection to identify the therapeutic targets. Through enrichment analyses and pharmacology network construction, we identified the core targets and the mechanism underlying the anti-AD function of *E. rutaecarpa*. We performed molecular docking and created a protein–ligand interaction profile to calculate the binding affinities, sites, and poses between the candidate ligands and the AD-related core targets, thus finding the novel lead molecules in *E. rutaecarpa* for the treatment of AD.

The design and flowchart of the entire study is presented below (Figure 1).

## 2. Results

### 2.1. Active Compounds of WZY Collection and Toxicity Study

The workflow of the entire study is shown in Figure 1. First, we obtained 202 compounds from databases and the literature, and screened these compounds with ADMET criterion. Finally, we collected 32 active compounds, as shown in Table 1. WZY1–22 were screened from the TCMSP database, and WZY23–32 were screened from the ETCM database. The selected 32 active compounds were composed of three categories: 27 alkaloids (including 11 indole alkaloids, 10 quinolone alkaloids, and 6 other alkaloids), three sterols, and two fatty acids (esters). In the toxicity study, we employed the ProTox-II server to determine the predicted toxicity of WZY active compounds, which included calculating the lethal dose 50 (LD50) and predictions of the toxicity class, hepatotoxicity, cytotoxicity, immunotoxicity, mutagenicity, and carcinogenicity. According to the data shown in Appendix A, the LD50 value ranged from 200 to 20,000; the LD50 class ranged from 3 to 6, with nine compounds in class 3, twenty-two compounds in class 4, and two compounds in class 6. In the prediction of hepatotoxicity, 31 of the compounds were inactive—all except for WZY22 (2-hydroxy-3-formyl-7-methoxycarbazole). In the prediction of several toxicity end points, 28 compounds were inactive with carcinogenicity, 15 compounds had inactive immunotoxicity, 11 compounds had inactive mutagenicity, and 30 compounds had inactive cytotoxicity. The toxicity prediction results showed that most of the WZY active compounds had good potential for further drug development.

### 2.2. Target Prediction

We used the TargetNet database to predict the potential target genes of these 32 active compounds. To obtain target genes with higher probability, we screened the target genes by setting a probability score cutoff ≥ 0.5. Finally, 229 target genes (after the repeated genes were deleted) with a high target probability were obtained (Appendix A). The compound–target network (Figure 2A) was constructed using Cytoscape (V3.8.0), and the size of nodes increased with the degree value counted by the analyzer app. As the network shows, alkaloids played the most important role: WZY28, WZY31, WZY32, and WZY24 were the most active compounds with the largest numbers of targets; and HTR1E, HTR5A, CA6, AHR, and DRD5 were the most targeted nodes.

### 2.3. Alzheimer’s Disease, the Disease in Which Core 229 Target Genes Are Mainly Enriched 

To detect the kinds of diseases these core 229 target genes are related to, we used Disease Ontology (DO) enrichment analyses. For ensuring the validity of the results, two kinds of analysis methods were employed. As shown in the bubble diagram (Figure 2B,C), the 229 target genes were mainly enriched in neuro-related diseases. Correspondingly, the TOP1 among the DO results was Alzheimer’s disease (Figure 2B). Another TOP1 was memory impairment among DisGeNET enrichment results (Figure 2C). The results of both methods of gene–disease association enrichment analyses indicated that WYZ active compounds might have an effect on Alzheimer’s disease.

### 2.4. Neural Construction, Activities, and Pathways That Core 229 Target Genes Primarily Enriched

All 229 target genes were analyzed by GO and KEGG pathway enrichment analyses, and the enriched results are displayed as bubble histograms or diagrams (Figure 3). GO biological processes were enriched mainly in the G-protein-coupled receptor-signaling pathway, synaptic signaling, cellular response to hormone stimuli, etc. (Figure 3A), and among these, synaptic signaling (Q = 1 × 10^−34.5^, enrich factor = 25%) was notable and corresponded to other enrichments. In addition, GO cellular component results primarily contained dendrites, post synapses, membrane rafts, and receptor complexes (Figure 3B). Furthermore, consistent with previous results, GO molecular function enrichment results mainly contained protein kinase activity, G-protein-coupled amine/peptide receptor activity, and bioactive lipid receptor activity (Figure 3C). Among these, postsynaptic neurotransmitter receptor activity (Q = 1 × 10^−9.3^ enrich factor = 5%) and tau protein binding (Q = 1 × 10^−8^ enrich factor = 4%) were the functions that were the most AD-related. Moreover, in KEGG enrichment analysis, the top 20 pathways are shown in Figure 3D. The primary pathways were also neuroactivity-related: neuroactive ligand–receptor interactions (Q = 1 × 10^−73.8^, gene number = 65), serotonergic synapses, sphingolipid-signaling pathways, dopaminergic synapses, cholinergic synapses, gap junctions, and steroid hormone biosynthesis. These pathways are highly related to AD. In conclusion, synapse signaling, hormones, and bioactive lipids were the main factors in compounds of WZY targeting AD.

### 2.5. AD Target Genes’ Collection, Compound–AD Intersection, and PPI Network Construction

We collected 9841 genes associated with AD from the Genecards and OMIM databases after removing repeated genes. Then, we filtered the number of AD genes to 1285 by setting an inference score cutoff ≥ 10. After that, a total of 87 genes were obtained by intersecting 1285 AD genes with 229 core genes targeted by the active compounds in WZY. A Venn diagram is shown in Figure 4A, and the detailed information of these 87 genes is displayed in Appendix A. For detecting these 87 genes’ interaction, a protein–protein interaction network was constructed using the STRING database. The PPI network of 87 genes contains 86 nodes and 607 edges, as shown in Figure 4C. Arranging the degree value of each node, we obtain the top nine nodes: SRC, TNF, PTGS2, ESR1, HSP90AA1, NOS3, NR3C1, AR, and SLC6A4, with their degree values, respectively, of: 40, 39, 34, 34, 33, 27, 27, 25, 24.

### 2.6. The 87 WZY–AD Intersection Genes Mainly Enriched in Alzheimer’s Disease

We further analyzed the Disease Ontology enrichment of 87 intersection genes, and the results showed that the 87 genes mainly enriched in Alzheimer’s disease (Q = 10^−27^, enrich factor = 51%) included those related with tauopathy, migraine, attention deficit disorder, hyperactivity disorder, synucleinopathy, anxiety disorder, mood disorder, etc. Obviously, the 87 intersection genes are mainly enriched in neural and mental diseases, especially AD and tauopathy (Figure 4B), which corresponds with the preliminary results shown in Figure 2B.

### 2.7. GO and KEGG Pathway Enrichment Analysis of 87 WZY–AD Intersection Genes

In addition, we conducted GO and KEGG pathway enrichment analysis to identify the enrichment information of these 87 WZY–AD intersection genes. The enriched results are displayed as bubble or histogram diagrams (Figure 5). GO biological processes (Figure 5A) were mainly enriched in neural processes. The five most significant terms, from top to bottom, were: cellular responses to organic cyclic compounds (Q = 1 × 10^−25.8^, enrich factor = 35.6%), responses to drugs, circulatory system processes, rhythmic processes, positive regulation of transferase activity, etc. In addition, GO cellular component (Figure 5B) results contained: post synapses (Q = 1 × 10^−12.6^, enrich factor = 24.1%), dendrites (Q = 1 × 10^−11.8^, enrich factor = 23%), and membrane rafts (Q = 1 × 10^−10.5^, enrich factor = 17.2%), followed by receptor complexes, perinuclear regions of cytoplasm, etc. Furthermore, consistent with previous results, GO molecular functions (Figure 5C) contained: neurotransmitter receptor activity, protein kinase activity, nuclear receptor activity, heme binding, protein kinase binding, tau protein binding, and hormone binding. Notably, neurotransmitter receptor activity (Q = 1 × 10^−13^, enrich factor = 15%) and tau protein binding (Q = 1 × 10^−6^, enrich factor = 7%) are highly AD-related. Moreover, in KEGG pathway enrichment analysis, the top 20 pathways are shown in Figure 5D. These were also neuroactivity-related, including: neuroactive ligand–receptor interaction, serotonergic synapses, pathways in cancer, dopaminergic synapses, gap junctions, etc.

### 2.8. Network Construction of Compounds Targeted to WZY–AD Intersection Genes 

As shown in Figure 5E, 119 nodes and 407 edges were found, and the network was built with 32 active compounds (V-shaped) targeting 87 AD–WZY intersection genes (round shape). The nodes’ color changes from green to yellow to red as the degree value increases from 1.0 to 26.0. The shape and size of the nodes also changes from small to large as the degree value increases. Among the 87 gene nodes, the most targeted gene was DRD5, followed by CASP9, PRKCB, TLR9, HTR3A, MAOA, HTR2C, MIF, CNR2, HTR1A, CNR1, etc. Nine core nodes with the highest degree values (from 87 intersection genes’ PPI network (Figure 4C)) are marked with a dark pink border, and their degree values in this C-T network (Figure 5E) ranged from 1 to 5. Among the 32 compounds, alkaloid compounds (V-shaped, marked by purple border) accounted for the majority, including the most targeted compounds: WZY28, WZY25, followed by WZY31, WZY27, WZY16, WZY19, WZY14, WZY29, WZY11, WZY11, and WZY3, with their degree value ranging from 26 to 8.

### 2.9. Molecular Docking

A total of 288 dockings were processed, and their binding affinity is shown in Figure 6. The molecular docking study, with its high accuracy, is a powerful tool for analyzing the interactions between active compounds and potential targets. To identify the functional effects of WZY active compounds in AD, the Python language and AutoDock Vina were used to detect the interaction between WZY active compounds and core AD-related targets. Nine core AD-related genes—SRC, TNF, PTGS2, ESR1, HSP90AA1, NOS3, NR3C1, AR, and SLC6A4—were the most important and interactive targets of the 32 WZY active compounds, as the previous network pharmacology results (Figure 4C) indicated. Three-dimensional structures of these nine macromolecular complexes were selected and downloaded from the PDB database. After removing their original ligands, they were docked with the 32 WZY active compounds in sequence. Additionally, the binding affinity of the original ligands of the complexes are shown in the bottom row of the table as positive controls. 

Lower binding affinity energy indicates a stronger binding affinity. The results showed that SRC (PDB ID 6E6E), TNF (PDB ID 6NH8), PTGS2 (PDB ID 5IKR), ESR1 (PDB ID 6VIG), HSP90AA1 (PDB ID 1YC1), NOS3 (PDB ID 7JRA), NR3C1 (PDB ID 6DXK), AR (PDB ID 5CJ6), and SLC6A4 (PDB ID 6VRH) all had good binding affinities with WZY active compounds. Furthermore, among these nine key target proteins, TNF, NOS3, SLC6A4, and PTGS2 tended to have best docking affinities with WZY active compounds. Among all 32 compounds, WZY21, WZY27, WZY11, WZY17, and WZY25 were the top five compounds with the best affinities toward the targeted macromolecules, with an average binding affinity lower than −9 kcal/mol. Notably, WZY27, WZY11, WZY17, and WZY25 are alkaloids, again validating that alkaloids of WZY play an important role in pharmacological functions. Further, compared with original ligands in the PTGS2 complex, WZY27, WZY17, WZY25, and WZY16 all had a lower binding affinity energy, suggesting that these active compounds have stronger binding affinities with the target protein PTGS2. In the docking list of AR, we found that WZY21, WZY27, WZY17, WZY25, and WZY16 had a stronger binding affinity than original ligands split from the PDB original complex. Also, in dockings of SLC6A4, AR, and SRC, we found a few binding compounds with stronger binding affinities than the positive controls (original ligands) from the PDB database, and these are marked with a purple asterisk.

The strongest dockings of each macromolecule (marked by purple-dotted squares in Figure 6) were chosen, and their binding sites, interactions, and distances are shown in Figure 7. Hydrogen bond interaction accounted for the largest proportion in different interactions of docking. All these ten dockings had hydrogen bond interactions, except for WZY27 (dehydroevodiamine)-SRC (6E6E) (Figure 7F) and WZY17 (dihydrorutaecarpine)-TNF (6NH8) (Figure 7J). Additionally, pi–cation interaction ranked second in different interactions of docking. Further, WZY14 (fordimine)-HSP90AA1 (1YC1) (Figure 7B), WZY21 (goshuyuamide II)-NOS3 (7JRA) (Figure 7E), WZY27 (dehydroevodiamine)-PTGS2 (5IKR) (Figure 7G), WZY16 (rutaecarpine)-SLC6A4 (6VRH) (Figure 7H), and WZY17 (dihydrorutaecarpine)-TNF (6NH8) (Figure 7J) all had pi–cation interactions.

As shown in Figure 6, four dockings had stronger binding affinities than the positive controls from the PDB database. These were (A) WZY25 (graveoline)-AR (5CJ6), (F) WZY27 (dehydroevodiamine)-SRC (6E6E), (G)WZY27 (dehydroevodiamine)-PTGS2 (5IKR), and (H) WZY16 (rutaecarpine)-SLC6A4 (6VRH). In these dockings, H-bonds, Van der Waal’s forces, and pi–cation interactions were observed.

### 2.10. Protein–Ligand Interaction Comparison of Potential Compounds versus Positive Controls

Based on the results of molecular docking and target prediction, we selected four potential compounds with the highest binding affinities and target prediction probabilities. These were: WZY25 (graveoline), WZY27 (dehydroevodiamine), WZY26 (5-methoxy-N, N-dimethyltryptamine), and WZY21 (goshuyuamide II). The corresponding targets of the four compounds are: AR (PDB ID 5CJ6), PTGS2 (PDB ID 6E6E), SLC6A4 (PDB ID 6VRH), and SRC (PDB ID 6E6E). The four targets’ 3D structures chosen from PDB all had original ligands. Respectively, these were: 51Y (2-chloro-4-{[(1R,2R)-2-hydroxy-2-methylcyclopentyl]amino}-3-methylbenzonitrile), mefenamic acid, paroxetine, and HVY (N-(2-chloro-6-methylphenyl)-2-[4-(4-methylpiperazin-1-yl)anilino]-4-[2-(propanoylamino)anilino]pyrimidine-5-carboxamide). The original ligands were used as positive controls. To compare the potential compounds with positive control ligands, we employed the Protein–Ligand Interaction Profiler (PLIP) server.

As shown in Figure 8A, WZY25 (graveoline) forms a closer π-stacking (perpendicular) interaction (4.8 Å, PHE-764) as well as five hydrophobic interactions (amino acid (AA) residues: LEU-701, LEU-704, MET-745, MET-749, PHE-876). The oxygen atom of its benzodioxole group forms a hydrogen bond (3.4 Å) with THR-877; 51Y (positive control) forms two hydrogen bonds, a π-stacking (perpendicular) interaction, and five hydrophobic interactions. 51Y is a selective androgen receptor modulator (SARM). It has an agonist activity at the human androgen receptor (EC50 = 0.000499 µM)[29]. The results showed that WZY25 (graveoline) is a potential lead molecule targeting AR.

As shown in Figure 8B, WZY27 (dehydroevodiamine) forms a hydrogen bond (3.8 Å) with the oxygen atom of the active site of SER-530 [30] as well as eleven hydrophobic interactions (4.8 Å), which is four more interactions than the mefenamic acid of the positive control. Mefenamic forms a hydrogen bond and seven hydrophobic interactions. Mefenamic acid is a non-steroidal anti-inflammatory agent with analgesic, anti-inflammatory, and antipyretic properties. It is an inhibitor of cyclooxygenase and has an inhibiting effect at PTGS2 (IC50 = 0.602 µM, bioassay record: AID 625244). The results showed that dehydroevodiamine is a potential lead molecule for inhibiting PTGS2.

Compared with paroxetine, WZY26 ((5-methoxy-N, N-dimethyltryptamine)) forms a shorter hydrogen bond (3.4 Å) with the oxygen atom of AA residue ASP-98 and a closer π-stacking (perpendicular) interaction (4.8 Å, TYR-176), but only one hydrophobic interaction (4.0 Å) with PHE-341 (Figure 8C). Paroxetine forms a hydrogen bond, a π-stacking (perpendicular) bond, and three hydrophobic interactions. Paroxetine is a selective serotonin-reuptake inhibitor (SSRI) drug. Its IC50 at SLC6A4 is 0.00004 µM [31].

Compared with HVY (N-(2-chloro-6-methylphenyl)-2-[4-(4-methylpiperazin-1-yl)anilino]-4-[2-(propanoylamino)anilino]pyrimidine-5-carboxamide), WZY21 forms three hydrogen bonds (3.3 Å, 3.6 Å, 3.1 Å) with AA residues: CYS-280, LYS-298, and THR-341, respectively. It forms a π–cation interaction (3.6 Å) with LYS-298 and eight hydrophobic bonds. HYY forms four hydrogen bonds and eight hydrophobic bonds but no π-cation interaction, and its IC50 at human SRC is 0.003 µM [32]. The results showed that WZY21 shows great potential as a SRC inhibitor.

## 3. Discussion

Our study found, firstly, that WZY active compounds primarily target proteins associated with AD and not just with migraines. Enrichment results further supported this finding. WZY compounds bind to AD node proteins with high affinity. A few of the compounds even displayed a higher affinity than the known effective drugs recorded in the PDB database, shown by the results of the batch molecular docking.

Among the almost two hundred compounds contained in WZY, we filtered and obtained 32 active compounds that had good BBB penetrability. Over 80% of them were alkaloids with various biological activities, including evodiamine and berberine, which have been widely studied for their beneficial functions in neural system diseases, especially AD [33,34].

However, most of these active compounds have not been studied clearly yet, and WZY has great untapped potential pharmaceutical value. In our study, we found that WZY mainly targets Alzheimer’s disease and memory impairment, followed by migraines, anxiety, depression, and mood disorders. Synaptic signaling and cellular response to hormone stimulus were the key targeted BPs. Neuron synapses, dendrites, axons, and membrane rafts were the main target locations. Moreover, the targets mainly enriched pathways: neuroactive ligand–receptor interaction and synapses (serotonergic, dopaminergic, and cholinergic) are deeply related to AD. From these results, we can conclude that WZY active compounds are highly affiliated with the neural system in aspects of cell components, functions, processes, and pathways. Working together, these compounds precisely target neural system diseases, especially AD.

In order to detect the effect and mechanism of WZY on AD, we collected and filtered the highly correlated genes of AD. These genes were then overlapped with 229 WZY target genes. Finally, we obtained 87 intersection genes. In the PPI network of intersection genes, we found that WZY might affect AD through inflammation and hormone node proteins. In the compounds–target network, we could see that the alkaloids of WZY played the most important role in targeting AD. Furthermore, we found that WZY impacts AD through the biological process of influencing cellular responses to organic cyclic compounds. This means the alkaloids of WZY play the most important role in producing pharmacological effects. In molecular functions, neurotransmitter receptor activity, tau binding, and hormone binding are disqualified. Thus, we speculated that WZY compounds might impact AD by influencing these functions. Finally, in pathways, WZY mainly affected AD through neuroactive ligand–receptor interactions, serotonergic synapses, dopaminergic synapses, cholinergic synapses, gap junctions, and inflammation.

Furthermore, we utilized the Python language to run batch Autodock Vina dockings to score the binding affinity of 32 WZY active compounds that bind to nine core node proteins. These were screened by the highest degree value of the PPI network of 87 intersection genes. In total, 288 compound–target dockings and nine original ligand–macromolecule dockings were processed. Based on the binding affinity energy results of Autodock Vina, lower binding affinity energy means a stronger binding affinity. We found that WZY active compounds all had a good binding affinity to the nine core proteins of AD. Alkaloids played the most effective role in targeting AD-related nodes, as the binding affinity of alkaloids was much stronger than other compounds contained in WZY. We could conclude that WZY compounds affect AD primarily through binding to node proteins of serotonergic synapse signaling (SLC6A4), hormones (PTGS2, ESR1, AR), inflammation (SRC, TNF, NOS3), transcription regulation (NR3C1), and molecular chaperones (HSP90AA1).

SLC6A4 (protein name: sodium-dependent serotonin transporter, SERT, 5HT transporter, 5HTT) plays the core role in serotonergic neurotransmission [35]. In a clinical trial, it was reported that AD+ patients might have lower SERT binding in limbic brain regions than AD-negative patients [36]. Increases in neuroinflammation and amyloid-beta 40 were associated with reduced SERT activity in a transgenic model of familial Alzheimer’s disease [37]. Another study reported that in human cortical and limbic areas with mild cognitive impairment, reduced SERT availability was observed [38]. Additionally, SLC6A4 is a typical and mature anti-depressant drug target; its gene polymorphism is associated with depression [39] and anxiety [40,41]. In our study, we found five compounds binding to SERT with a higher binding affinity than the positive control (original ligands) from the PDB database. This means that these compounds might contain a good medical promise in protecting against AD, anxiety, and depression by binding to SERT.

SRC (protein name: proto-oncogene tyrosine-protein kinase Src, Src) and TNF (protein name: tumor necrosis factor, TNF) are key proteins in AD pathological development. Inhibition of Src kinase activity attenuates amyloid-associated microgliosis in AD transgenic mouse models [42]. Impaired Src signaling involves GluN2B-composed NMDARs and post-synaptic actin cytoskeleton depolymerization in the hippocampus in early stages of AD [43]. Src activation in microglia have been shown to mediate TNF production. Inhibiting Src in microglia RhoA-deficient mice attenuates microglia dysregulation [44]. Possession of the TNF-alpha T allele significantly increases the risk of vascular dementia and increases the risk of Alzheimer’s disease associated with apolipoprotein E (APOE) [45]. In our study, we found nine compounds that bound to Src with a higher binding affinity than the positive control (original ligands) from the PDB database. This shows the great anti-neuroinflammation potential of these compounds in treating AD.

PTGS2 (protein name: prostaglandin G/H synthase 2, cyclooxygenase-2, COX-2) is the target of nonsteroidal anti-inflammatory drugs (NSAIDs) such as aspirin and ibuprofen [46]. Long-term use of these drugs reduces fatal thrombotic events and the development of Alzheimer’s disease. In Chinese traditional medicine, WZY is a traditional drug for migraines, possibly relating to alkaloids’ strong affinity to PTGS2, which corresponds with our finding—four compounds contained in WZY showed great value in treating migraines. These had a higher binding affinity to PTGS2 than the original ligand (clinical drug) from the PDB database.

ESR1 (protein name: estrogen receptor, ER) and AR (protein name: androgen receptor, AR) play core roles in the AD pathology of hormone modulation. In addition, they are the targets of many widely used clinical drugs in the treatment of migraines [46,47]. In AD, both ER and AR take part in numerous protective actions attenuating multiple aspects of AD-related neuropathology [48]. Estrogen’s cognitive-enhancing effect is impaired as age advances, probably related to age-linked alterations in ERs’ signaling and expression [49]. Further, the estrogen deficiency in menopausal women was supposed to be tightly linked with AD [50]. Similarly, experimental research indicated a protective effect in treating AD by hormone replacement therapy [51]. Estrogen functions as a neuroprotective agent for protecting against glutamate toxicities and decreasing amyloid beta (Aβ) [52], enhancing synaptic plasticity, maintaining neurotrophic components, assisting transcription factor initiation, reducing inflammation in the brain [53,54], and decreasing tau protein hyperphosphorylation [55,56,57]. Correspondingly, the function of estrogen is generally regulated by ER [51]. In addition, novel hormone preparations or SERMs (selective estrogen receptor modulators) that allow for brain-specific effects in the treatment of AD [58] may be developed.

AR participates in modulating AD pathology, as indicated by clinical trials and animal experiments. Men diagnosed with dementia have demonstrated reduced levels of testosterone, including prior to the onset of dementia [59]. Among men with AD, testosterone supplementation has been shown to improve spatial and verbal memory [60,61,62]. Androgen depletion is implicated in the development of Alzheimer’s disease, as circulating testosterone levels in older men are inversely correlated with levels of amyloid β (Aβ) protein in the brain [63]. SARMs (selective androgen receptor modulators) are being explored as a potential therapy for Alzheimer’s disease [64]. Excitingly, we found several new alkaloids with strong binding affinities that exhibit great potential to be developed as novel SARMs.

HSP90AA1 (protein name: heat shock protein HSP 90-alpha, HSP90α) and NR3C1 (protein name: glucocorticoid receptors, GRs) play important roles in AD pathology and are popular therapeutic targets in the research. In AD, HSP90 inhibitors may redirect neuronal aggregate formation and protect against protein toxicity by the activation of HSF-1 and the subsequent induction of heat shock proteins, such as Hsp70 [65]. Inhibition of HSP90 leads to a decrease in the formation of tau and Aβ aggregate, as well as the degradation of the tau phosphorylating kinases [66]. The development of inhibitors specifically targeting HSP90 or its interaction with Aha1 might be an efficient way to control tau aggregation with limited side effects [67]. There are a lot of data collected so far indicating that HSP90 may serve as a prospective target for pharmacological intervention in neurodegenerative diseases [66].

GRs modulated dendritic spine plasticity and microglia activity in an animal model of Alzheimer’s disease. Agonist treatment of GRs reduced dendritic spine density and induced the proliferation and activation of microglia. Antagonist treatment of GRs enhanced dendritic spine density, decreased microglia density, and improved behavioral performance [68]. GRs seem to occupy a central position in the pathophysiology of AD, and animal experiments have indicated that impairment of GR signaling potentiates amyloid-β oligomer-induced pathology. Additionally, treatment with a kind of selective GR modulator (sGRm) normalized plasma GC levels, and all behavioral and biochemical parameters were also analyzed [69].

It is worth mentioning that several compounds showed a stronger binding affinity to SERT, SRC, PTGS2, and AR than the original ligands (positive controls) split from complexes that are recorded as mature or developing clinical drugs in the PDB database, especially WZY25 (graveoline), WZY27 (dehydroevodiamine), and WZY21 (goshuyuamide II). In addition, when the QSAR target prediction probability was taken into consideration, WZY26 (5-Methoxy-N, N-dimethyltryptamine) had a high probability, but its binding affinity was not the highest.

The indole alkaloids (WZY21, WZY26, and WZY27) show great potential to be lead molecules. Their nitrogen heterocyclic ring plays the most important role in protein–ligand interaction. The nitrogen atom of the ring forms hydrogen bonds with -OH or -NH3 of the AA residues. The oxygen atom of the oxygen heterocyclic ring in WZY25 plays the role of forming a hydrogen bond with the -OH of the AA residues. Overall, the nitrogen and oxygen heterocyclic rings are highly active, perform an important function in protein–ligand interactions, and play an important role in exerting pharmacological activity. This is a new finding that shows a promising prospect for drug development regarding aspects of single compounds or multiple selected compounds of WZY in the treatment of AD. Considering the aspect of the mechanism involved, WZY is a hopeful drug for treating AD through the multiple-target approach. It combines the modulation of serotonergic synapse signaling, anti-inflammation, and hormone modulation. It is a potential effective therapy that could be widely developed and applied in clinics for its natural characteristics and its advantages in the homology of food and medicine.

In ancient China, there was no definition for or disease name of AD in Chinese traditional medicine. This may be an important reason for the gap in WZY active compounds and AD treatment. We speculate that in ancient China, shorter average lives accompanied with a lower incidence of AD might account for the reason that the medicinal value of WZY for AD treatment has not been utilized. Our research, using advanced bioinformation technologies, digs out the untapped medicinal value of the traditional Chinese herb WZY. This means that there might be more herbs with unknown potential medical value waiting to be discovered. Traditional Chinese medicine has shown rapid innovations in recent decades worldwide, and our study provides a new idea for the research and development of traditional Chinese medicine in the future.

## 4. Materials and Methods

### 4.1. Compound Collection, ADMET Screening, and Toxicity Prediction

From the Traditional Chinese Medicine System Pharmacology (TCMSP; https://pubchem.ncbi.nlm.nih.gov/, accessed on 29 September 2022) database [70], we collected the basic information of WZY compounds: compound names, PubChem CID, canonical SMILES, MF, and MW. Drug likeness (DL), oral bioavailability (OB), and blood–brain barrier (BBB) information was collected. The criteria for distribution, metabolism, excretion, and toxicity (ADMET) were: DL ≥ 0.18, OB ≥ 30%, and BBB ≥ −0.3. We also collected compounds from the Encyclopedia of Traditional Chinese Medicine (ETCM; https://pubchem.ncbi.nlm.nih.gov/, accessed on 29 September 2022) database [71] for Appendix A. In the ETCM database, the ADMET criteria included: drug likeness was graded at good or moderate; ADMET BBB level was at 0, 1, or 2. The obtained compounds were submitted to the PubChem (https://pubchem.ncbi.nlm.nih.gov/, accessed on 3 November 2022) database for further confirmation. Finally, these compounds from the TCMSP and ETCM databases were summarized, and the repeated ones were removed. We employed the ProTox-II (https://ox-new.charite.de/protox_II/, accessed on 25 June 2023) server to determine the predicted toxicity of WZY active compounds. The SMILES of compounds were submitted online, and the lethal dose 50 (LD50), prediction of toxicity class, hepatotoxicity, and toxicity end points (cytotoxicity, immunotoxicity, mutagenicity, and carcinogenicity) were calculated.

### 4.2. Target Prediction

Quantitative structure–activity relationships: The TargetNet (QSAR-TargetNet) webserver [72] was used to predict the potential targets of active compounds of WZY. The parameter for prediction was AUC ≥ 0.75; the fingerprint type was ECFP4. The TargetNet webserver scored drug–target interaction probability from 0 to 1 for each prediction. After gathering all the information for the predicted targets, including the UniProt ID, protein name, and probability score, we filtered the targets by setting a cutoff of probability ≥ 0.5. Then, the UniProt database [73] was employed to translate the UniProt ID to a gene symbol and gene ID. At the same time, the species of genes was checked, and non-human genes were eliminated.

### 4.3. PPI Networking

The STRING database [74] was used to obtain protein–protein interaction (PPI) data, and Cytoscape (V3.8.0) [75] was used to visualize the network. Then, we employed the network analyzer app to analyze the PPI network to obtain information including the degree value and the numbers of nodes and edges. The MCODE plugin was used to find densely connected clusters, and the nodes were visualized by graduated color changing with MCODE scores.

### 4.4. Disease Ontology Enrichment Analysis

Disease ontology analysis can detect the kinds of diseases with which the genes targeted by the compounds are mainly related. The R language with the DOSE package [76] and the DisGeNET [77] assay of the Metascape database [78] were used to analyze the disease ontology enrichment of compounds’ targeted genes. For visualizing the enrichment results, bubble plots were drawn using the R language and the ggplot2 package.

### 4.5. Gene Ontology (GO) and Kyoto Encyclopedia of Genes and Genomes (KEGG) Enrichment Analysis

GO [79] enrichment, including GO biological processes, GO molecular functions, GO cellular components, and KEGG pathway [80] enrichment, were analyzed by the Metascape database [78]. The parameters were: min overlap ≥ 3, *p* value ≤ 0.01, min enrichment ≥ 1.5. The results were then visualized by bubble and bar plots using the R language and the ggplot2 package.

### 4.6. Compound–Target Networking

Compound–target network data generated from target predictions (see Section 4.2) were visualized with Cytoscape (V 3.8.0) [75], and the network analyzer plugin was used to analyze the network.

### 4.7. AD Target Screening and Identification of Target–AD Intersection Genes

AD targets were collected from the GeneCards database [81], the Comparative Toxicogenomic Database (CTD) [82], and the Online Mendelian Inheritance in Man (OMIM) [83] database. After removing repeated genes, we received the total collection of targets. Consequently, the targets from the Genecards database contained the total collection. Thus, we filtered the AD targets by one standard: the relevance score generated from the Genecards database. The cutoff of the relevance score was greater than or equal to 10.

### 4.8. Molecular Docking

Three-dimensional structures of macromolecules were selected and downloaded from the Protein Data Bank (PDB) in PDB format. All of these were crystal complexes with particular ligands, including: SRC (PDB ID 6E6E) [84], TNF (PDB ID 6NH8) [85], PTGS2 (PDB ID 5IKR) [30], ESR1 (PDB ID 6VIG) [86], HSP90AA1 (PDB ID 1YC1) [87], NOS3 (PDB ID 7JRA) [88], NR3C1 (PDB ID 6DXK) [89], AR (PDB ID 5CJ6) [29], and SLC6A4 (PDB ID 6VRH) [90]. Their resolutions, respectively, were: 2.15 Å, 1.80 Å, 2.34 Å, 1.45 Å, 1.70 Å, 2.10 Å, 3.05 Å, 2.07 Å, and 3.30 Å. Three-dimensional structures of WZY active compounds were downloaded from different databases, MOL2 format structures of WZY 1–22 were directly downloaded from the TCMSP database, and SDF format structures of WZY23–32 were download from PubChem.

For macromolecule preparation, in PDB format, the original ligands were split from complexes, and water and polar hydrogens were deleted using PyMoL (2.0). Then, AutoDockTools-1.5.6 [82] was employed to convert the PDB format to PDBQT. For ligand preparation, the MOL2 and SDF formats were uniformly converted to PDB format in PyMOL (2.0), and the polar hydrogens of ligands were deleted. Then, the PDB format was converted to PDBQT in Au toDockTools-1.5.6. Further, the original ligands split from macromolecule complexes were prepared in the same way as the positive control group rather than that of WZY active compounds. AutoDockTools-1.5.6 [91] was employed to create the grid box and config data of macromolecules and compounds for the next docking. Then, Autodock Vina [92] was operated to calculate the binding affinities, residue sites, bonds, and other interactions of each docking.

### 4.9. Protein–Ligand Interaction Profiler (PLIP)

The docking results of ligands’ poses were split, and the top pose was prepared in PDB format. Then, the ligands’ PDB format was combined with the corresponding PDB format of the macromolecule (water, nonpolar hydrogen atoms, and ligands were removed) and saved as a complex. The complexes were submitted to the Protein–Ligand Interaction Profiler (PLIP, https://plip-tool.biotec.tu-dresden.de/plip-web/plip/index, accessed on 21 June 2023) webserver, and all the calculated results were saved for further analyses and display.

## 5. Conclusions

In silico techniques—compound collection, target prediction, enrichment analyses, pharmacology network construction, hub node identification, and molecular docking—were used to confirm and elucidate the mechanism underlying the anti-AD effects of *E. rutaecarpa* in this work. Meanwhile, standard ADME screening, QSAR-based target prediction, and toxicity studies were utilized to confirm the relevant characteristics of compounds. Then, the predicted targets were overlapped with a collection of AD-related targets to identify the therapeutic targets, and through enrichment analyses and pharmacology network construction, we identified the core targets and the mechanism underlying the anti-AD function of *E. rutaecarpa*. Molecular docking and protein–ligand interaction profiling were performed to calculate the binding affinities, sites, and poses between the candidate ligands and the AD-related core targets. This explained the underlying mechanism for treating AD and helped to find the novel lead molecules in *E. rutaecarpa* for the treatment of AD. Based on the results of this study, the main conclusions can be summarized as follows:

WZY showed significant AD-targeting characteristics from the result of QSAR target prediction. Alkaloids of WZY played the main role in targeting AD-related genes.

WZY affected the pathologic development of AD through serotonergic synapse signaling, hormones, anti-neuroinflammation, transcription regulation, and molecular chaperone pathways.

HSP90AA1, PTGS2, ESR1, AR, SRC, TNF, NOS3, and NR3C1 were identified as the core targets of WZY active compounds targeting AD through PPI network construction.

Molecular docking results indicated that in the dockings with PTGS2, AR, SLCA64, and SRC, alkaloids, especially indole alkaloids, showed stronger binding affinities than some known clinical or preclinical small-molecule drugs.

The protein–ligand interaction results showed that the nitrogen atoms in the nitrogen heterocyclic ring of indole alkaloids (WZY26 (graveoline), WZY27 (dehydroevodiamine), and WZY21 (goshuyuamide II)), as well the oxygen atom in the oxygen heterocyclic ring of other alkaloids (WZY25 (graveoline)), played the most important role in forming hydrogen bonds with the nitrogen or oxygen atoms of AA residues of AD-related targets. The benzene ring of alkaloids played the second most important role, as π-stacking and π–cation interactions with the benzene ring of AA residues could be formed. Graveoline (WZY25), 5-methoxy-N, N-dimethyltryptamine (WZY26), dehydroevodiamine (WZY27), and goshuyuamide II (WZY21) were identified as lead molecules for targeting AD-related core targets (AR, SLC6A4, PTGS2, and SRC, respectively).

It is a new finding that WZY is a hopeful drug for treating AD through single compounds or multiple compounds of WZY. In the future, based on the pathways and core nodes discovered here, we will conduct in vivo and in vitro experiments to verify the effects of the identified lead molecules on AD.

## Figures and Tables

**Figure 1 molecules-28-05846-f001:**
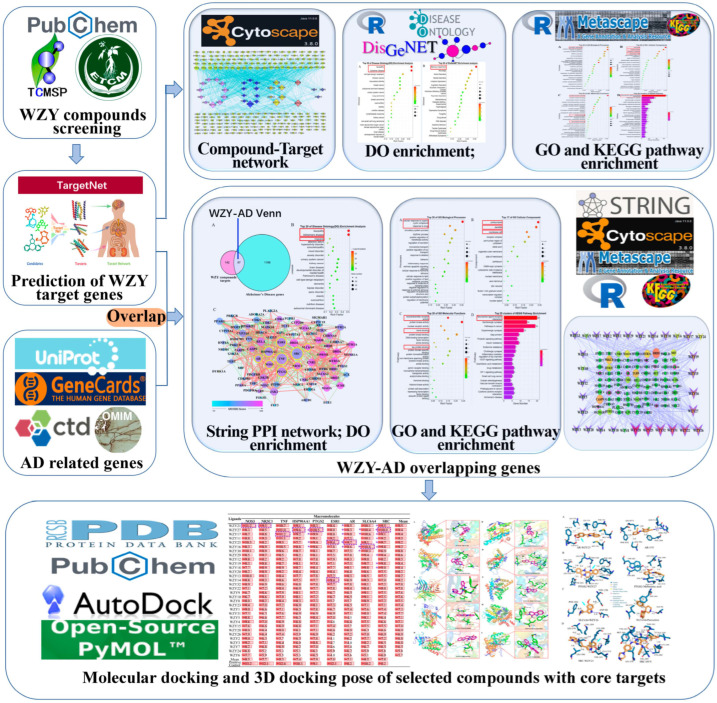
Workflow scheme. Our study was composed of three main steps. The first step consisted of compound screening and target prediction, then DO enrichment to focus on the targeted disease. The second step consisted of AD-related gene collection and overlapping with compound target genes, followed by a PPI network, GO and KEGG pathway enrichment, and C-T (overlapped) network construction, to find the core node proteins and pathways targeting AD. The third step consisted of molecular docking to further identify the underlying mechanism of WZY compounds in treating AD.

**Figure 2 molecules-28-05846-f002:**
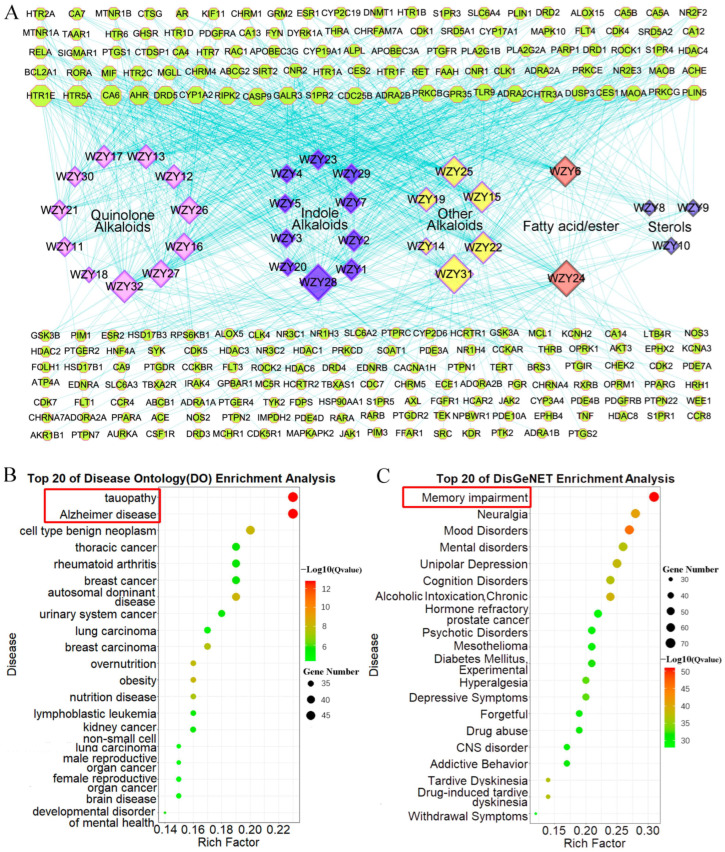
Compound-target network of 32 active compounds and bubble plot of DO enrichment analysis results of 229 target genes. (**A**) Compound–target network of 32 active compounds and target genes of WZY. The diamond nodes represent the compounds (pink for quinolone alkaloids, blue for indole alkaloids, yellow for other alkaloids, orange for fatty acids/esters, purple for sterols). The green octagon nodes with a red border represent the target genes. The nodes’ size changes from small to large as their degree value increases. (**B**) Bubble plot of top 20 DO enrichment analysis results of 229 WZY target genes, 1 × 10^−12.8^ < Q value < 1 × 10^−4.3^. The notable diseases are marked by red boxes. (**C**) Bubble plot of top 20 DisGeNET enrichment results of 229 WZY target genes, 1 × 10^−51^ < Q value < 1 × 10^−28^.

**Figure 3 molecules-28-05846-f003:**
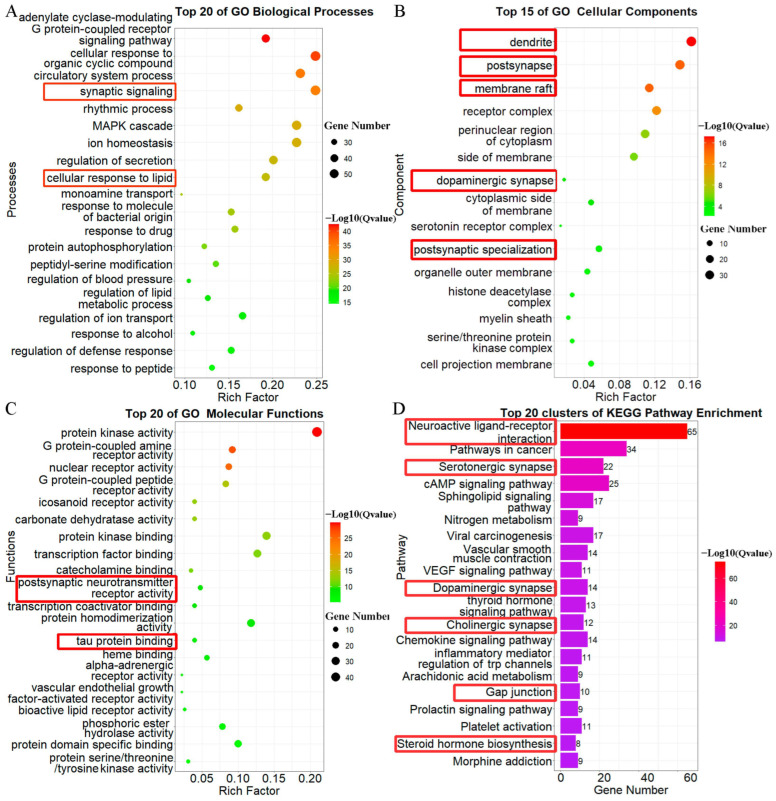
Bubble plots of GO terms and bar plot of KEGG pathway enrichment analysis of the 229 target genes. The AD-related and notable terms and pathways are marked by red boxes. (**A**) Top 20 significant terms of biological processes’ enrichment result, 1 × 10^−42.3^ < Q value < 1 × 10^−14.4^. (**B**) Top 15 significant terms of cellular components’ enrichment result, 1 × 10^−17.2^ < Q value< 1 × 10^−2.25^. (**C**) Top 20 significant terms of molecular functions’ enrichment result, 1 × 10^−29.8^ < Q value< 1 × 10^−5.3^. (**D**) Top 20 significant terms of KEGG pathway enrichment result, 1 × 10^−73.8^ < Q value < 1 × 10^−6.2^.

**Figure 4 molecules-28-05846-f004:**
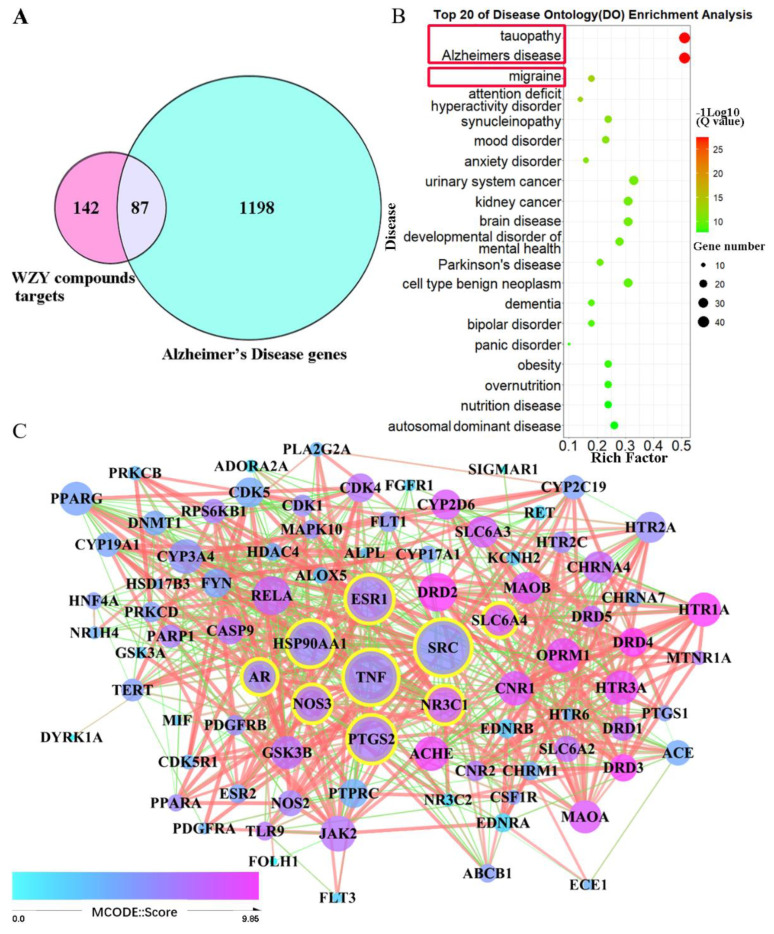
Venn, DO, and PPI plots of 87 intersection genes. (**A**) Venn diagram of WZY active compounds overlapping with AD-related genes. (**B**) Bubble plot of top 20 significant terms of DO enrichment of 87 overlapping genes, 1 × 10^−27.3^< Q value <1 × 10^−7.8^, the AD-related and notable terms and pathways are marked by red boxes. (**C**) PPI network of 87 overlapping target genes. Clusters with different MCODE scores are represented by different colors, as the legend shows. The size of the node is larger as its degree value increases. The 9 nodes with the highest degree values are represented by a yellow border.

**Figure 5 molecules-28-05846-f005:**
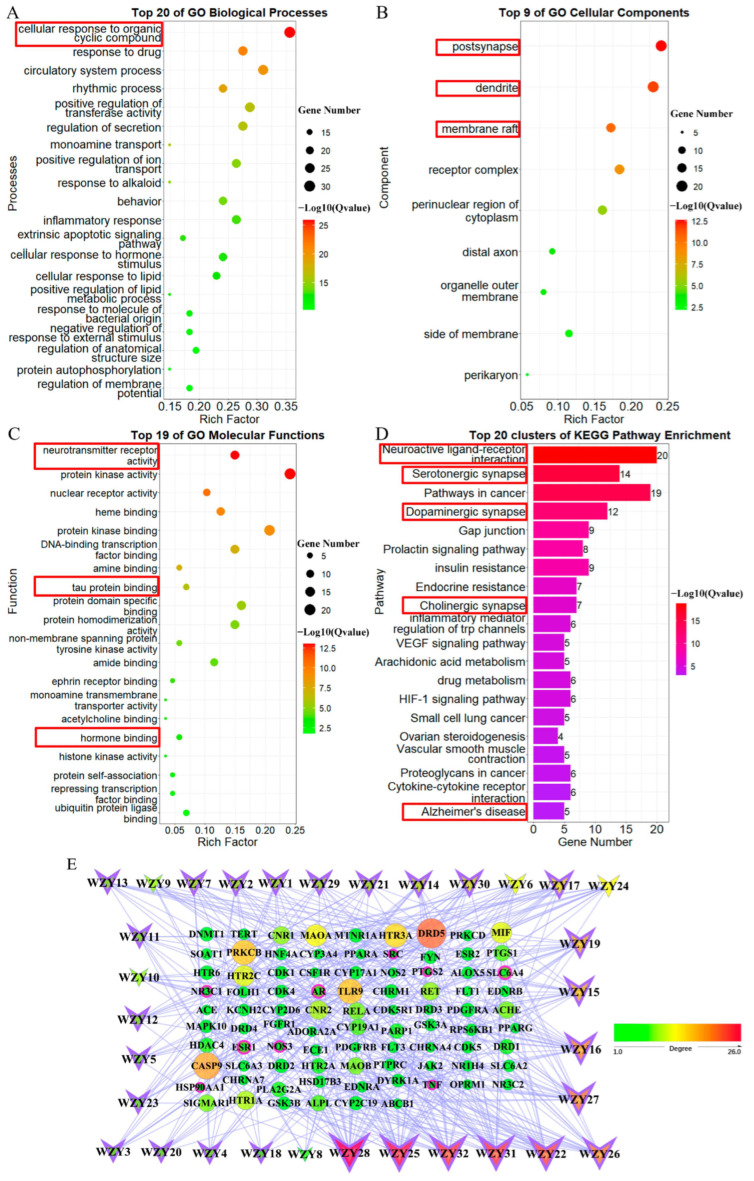
GO and KEGG pathway enrichment analyses results and compound–target network of 87 intersection genes. The AD-related and notable terms and pathways are marked by red boxes. (**A**) Top 20 significant terms of biological processes’ enrichment result, 1 × 10^−25.8^ < Q value < 1 × 10^−10.3^. (**B**) Top 9 significant terms of cellular components’ enrichment result, 1 × 10^−12.6^ < Q value < 1 × 10^−2.21^. (**C**) Top 19 significant terms of molecular functions’ enrichment result, 1 × 10^−13^ < Q value < 1 × 10^−2.36^. (**D**) Top 20 significant terms of KEGG pathway enrichment result, 1 × 10^−18.7^ < Q value < 1 × 10^−3^. (**E**) Compound–target network of 32 WZY active compounds targeting 87 intersection genes. The “V” nodes represent compounds (purple border for alkaloids). The round nodes represent target genes (pink border for 9 core AD-related genes). The size of the node is larger as its degree value increases. As the legend shows, the color of nodes changes from green to yellow to red along with their degree value.

**Figure 6 molecules-28-05846-f006:**
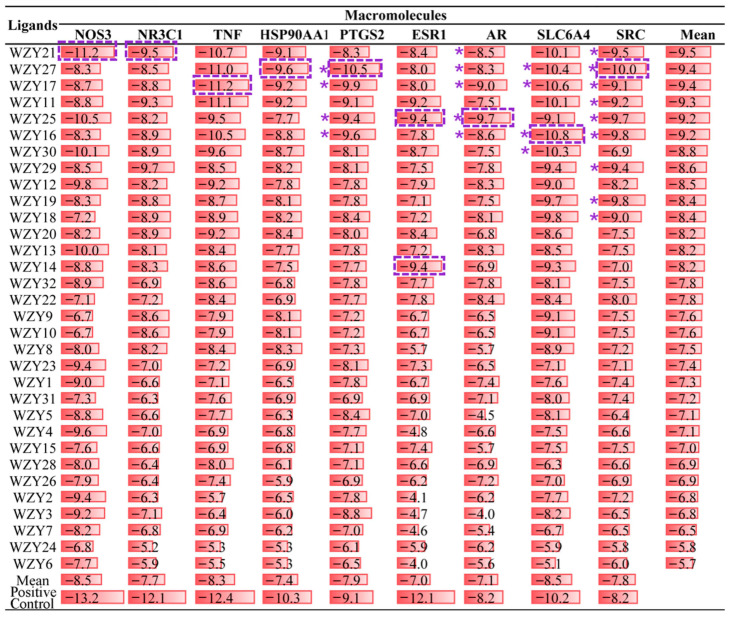
Scheme of binding energy (kcal/mol) of molecular dockings of WZY active compounds binding to 9 key node proteins of AD. For each macromolecule, the compound with lowest binding affinity energy (strongest binding affinity) is marked with a purple-dotted square border. Dockings with lower binding affinity energy than the positive control are marked with purple “*”.

**Figure 7 molecules-28-05846-f007:**
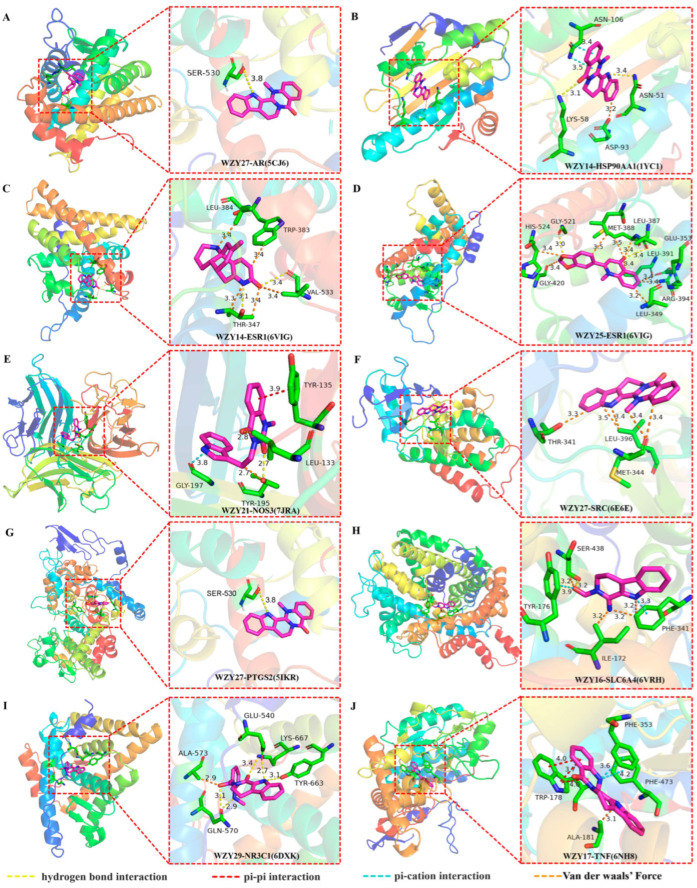
Molecular docking analysis of 9 core AD-related proteins bound to compounds with highest affinities (correspond to dockings with purple-dotted squares shown in Figure 6). (**A**) Sites of WZY25 (graveoline) binding to AR (PDB ID 5CJ6). (**B**) Sites of WZY14 (fordimine) binding to HSP90AA1 (PDB ID 1YC1). (**C**) Sites of WZY14 (fordimine) binding to ESR1 (PDB ID 6VIG). (**D**) Sites of WZY25 (graveoline) binding to ESR1 (PDB ID 6VIG). (**E**) Sites of WZY21 (goshuyuamide II) binding to NOS3 (PDB ID 7JRA). (**F**) Sites of WZY27 (dehydroevodiamine) binding to SRC (PDB ID 6E6E). (**G**) Sites of WZY27 (dehydroevodiamine) binding to PTGS2 (PDB ID 6E6E). (**H**) Sites of WZY16 (rutaecarpine) binding to SLC6A4 (PDB ID 6VRH). (**I**) Sites of WZY29 (rhetsinine) binding to NR3C1 (PDB ID 6DXK). (**J**) Sites of WZY17 (dihydrorutaecarpine) binding to TNF (PDB ID 6NH8).

**Figure 8 molecules-28-05846-f008:**
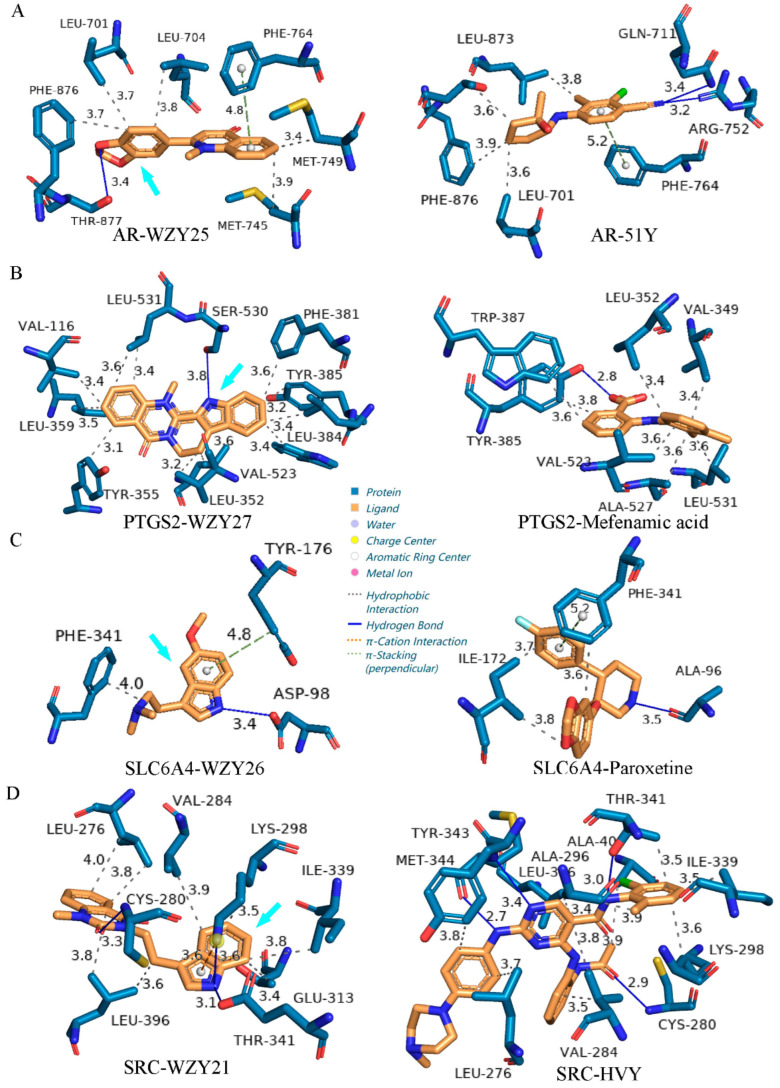
Binding sites’ comparison of potential compounds versus positive control ligands. (**A**) Binding site comparison of WZY25 (graveoline) versus 51Y (positive control) in docking with AR (PDB ID 5CJ6). (**B**) Binding site comparison of WZY27 (dehydroevodiamine) versus mefenamic acid (positive control) in docking with PTGS2 (PDB ID 5IKR). (**C**) Binding site comparison of WZY26 (dehydroevodiamine) versus paroxetine (positive control) in docking with SLC6A4 (PDB ID 6VRH). (**D**) Binding site comparison of WZY26 (5-methoxy-N, N-dimethyltryptamine) versus HVY (positive control) in docking with SRC (PDB ID 5IKR).

**Table 1 molecules-28-05846-t001:** Classification of 32 active compounds in *E. rutaecarpa*.

Code	Compound Name	PubChem CID	MW
Quinolone Alkaloids			
WZY1	1-methyl-2-nonyl-4-quinolone	13967189	285.47
WZY2	1-methyl-2-undecyl-4-quinolone	5319811	313.53
WZY3	Evocarpine	5317303	339.57
WZY4	1-methyl-2-[(Z)-undec-6-enyl]-4-quinolone	5319810	311.51
WZY5	1-methyl-2-[(Z)-pentadec-10-enyl]-4-quinolone	5319752	367.63
WZY7	1-methyl-2-pentadecyl-4-quinolone	5319753	369.65
WZY20	Hydroxyevodiamine	71307457	319.39
WZY23	1-methyl-2-[(Z)-5-undecenyl]-4(1*H*)-quinolone	5319809	311.46
WZY28	Echinopsine	6748	159.18
WZY29	Rhetsinine	99652	319.36
Indole Alkaloids			
WZY11	Evodiamine	442088	303.39
WZY12	Evodiamide	189454	307.43
WZY13	N-(2-methylaminobenzoyl) tryptamine	5319506	293.4
WZY16	Rutaecarpine	65752	287.34
WZY17	Dihydrorutaecarpine	N/A	289.36
WZY18	Goshuyuamide I	5317827	305.41
WZY21	Goshuyuamide II	5317828	319.39
WZY26	5-methoxy-N, N-dimethyltryptamine	1832	218.29
WZY27	Dehydroevodiamine	9817839	301.34
WZY30	14-formyldihydrorutaecarpine	5317369	317.34
WZY32	l-oxonoreleagnine	87371	186.21
Other Alkaloids			
WZY14	Fordimine	5462442	256.38
WZY15	Evodione	624052	292.36
WZY19	Berberine	2353	336.39
WZY22	2-hydroxy-3-formyl-7-methoxycarbazole	189687	241.26
WZY25	Graveoline	353825	279.29
WZY31	Kokusaginine	10227	259.26
Fatty acid/ester			
WZY6	Icosa-11,14,17-trienoic acid methyl ester	5367326	320.57
WZY24	Goshuyic acid	5312409	224.34
Sterols			
WZY8	24-methyl-31-norlanost-9(11)-enol	5319735	428.82
WZY9	Beta-sitosterol	86821	414.79
WZY10	Sitosterol	222284	414.79

Note: WZY1–WZY22 were screened from the TCMSP database; WZY23–WZY32 were screened from the ETCM database.

## Data Availability

The data presented in this study are available on request from the corresponding author.

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
