# Peer review of "Elucidation of Pharmacological Mechanism Underlying the Anti-Alzheimer’s Disease Effects of Evodia rutaecarpa and Discovery of Novel Lead Molecules: An In Silico Study"

_molecules, 2023, doi:10.3390/molecules28155846_

Round 1

Reviewer 1 Report

Title;Pharmacological Mechanism Underlying the Anti-Alzheimer’s Disease Effects of Evodia rutaecarpa and In Silico Discovery of Novel Lead Molecules
Comments;In my view, the results obtained in this study are worthy for publication. The manuscript needs major essential revision before publication. I would like to overview the revised version of the manuscript. I have the following comments/suggestions for authors to address before final decision on the manuscript.
1. Authors have written, "we analyzed the molecular dynamics’ trajectories of them, thus to find the novel lead molecules in E. rutaecarpa for the treatment of AD." but there is no MD simulations data provided in the manuscript.
2. Authors have written 288 docking of WZY compounds in the flow chart, which is need to be corrected.
3. Authors have written "The workflow of our whole study was shown in Figure 1." Remove "our" from the sentence.
4. Rewrite the table legend "Table 1. Screened active compounds of Wuzhuyu (32 compounds)."
5. Authors have advised to redraw the figures to increase the visibility of the text written in the images.
6. “role in binding the key nodes of AD;”: Binding the key nodes OR binding on the key nodes. Authors should frame the sentence very carefully.
7. The last sentence of the Abstract runs for almost 8 lines. It is a considerably long sentence and authors should break it down into smaller segments.
8. In the Introduction section the author should refer to the research paper and comment on recent in-silico techniques. It will be good information for the readers. I would like to recommend several papers, among many others, providing further explanation on this topic: PMID: 35276295 PMID: 23772360 PMID: 18367244 PMID: 33465692 PMID: 31133639 PMID: 31138032
9. Figure 1: The title provided for the last section “288…..macromolecules” should be reframed.
10. Consider changing “Active compounds of WZY and predicting target genes screening” to “Active compounds of WZY and screening target genes”.
11. Figure 4: The very first word of the figure legend is spelled wrongly.
12. “Alzheimer’s Disease, 87 WZY-AD intersection genes primarily associated with”: Incomplete heading. Re-write.
13. Section 4.8. The details are missing. Provide PDB Ids of all the crystal structures along with citations and their respective resolutions.
14. In methodology authors must have to elaborate more on the process of selecting and preparing the 3D structures of the macromolecular complexes from the PDB database for docking with the WZY active compounds. How were the original ligands removed, and were any additional steps taken to ensure the reliability of the docked structures?
15. What were the criteria or thresholds used to determine the importance and interactiveness of the identified targets? Were these based on binding affinity values or other factors?
16. In Figure 7 (h) was there any specific reason for not highlighting the sites of WZY16 (rutaecarpine) binding to SLC6A4 (PDB ID 6VRH)?
17. What specific methods or criteria were used to identify Evodia rutaecarpa (Wuzhuyu) as a potential drug for Alzheimer's Disease (AD) treatment? Were there any previous studies or evidence that led to this investigation?
18. What specific findings or observations were made regarding the binding of E. rutaecarpa compounds to the key nodes? Were there any compounds that demonstrated notable or exceptional binding affinities?
19. The authors mentioned that they included a conclusion section to provide a simplified overview of the complex results discussed. However, the current conclusion remains confusing. Therefore, it would be helpful to rewrite or incorporate the major findings more clearly in the conclusion section.
20“hub nodes identification and molecular docking techniques” Authors have used only a single molecular technique. So it's better to write technique instead of techniques.
21. “furthermore, we analyzed the molecular dynamics’ trajectories of them,” Authors have not performed any MD simulations in the study. It is not worthy to write it until or unless it is performed.
22. “TCMSP database and WZY23–32 were screened from ETCM database.” Mention the proper names of both the databases used here.
23. “pathway, secretion, immune, cellular response to hormone stimulus et al.” Mentioning about secretion, immune is not clear enough in the statement. Also it is not appropriate to use et al. in the statement.
24. In section 2.3 authors have used “et al.” too much. It does not seem good. Authors should find some other way of writing.
25. “factor = 51%), tauopathy, margarine,” Misspelled word in the line. It must be migraine instead of margarine

Minor editing of English language required

Reviewer 2 Report

Authors have presented an interesting manuscript entitled “Pharmacological Mechanism Underlying the Anti-Alzheimer’s Disease Effects of Evodia rutaecarpa and In Silico Discovery of Novel Lead MoleculesThe article is quite interesting and explored the new target of alkaloids in WZY which seem to have a great potential to be developed as an effective drug for treating Alzheimer’s disease. The study design and the applied experiment are good. The authors explored the mechanism of the active compounds of WZY using pharmacology network and molecular docking. In my opinion, the manuscript can be published in Molecules. Anyway, my suggestion, in vitro/ in vivo model should be confirm in the future.

Author Response

Dear Reviewer,

Thank you for your time and patience in reviewing this manuscript, and we are very appreciated your positive comments on the merits. We are really grateful for your recognition of our work. Thank you! best wishes for you!

Yours sincerely,

The Authors

Reviewer 3 Report

Alzheimer’s Disease is one of the first-line diseases and is of primary interest. The current manuscript entitled "Pharmacological Mechanism Underlying the Anti-Alzheimer’s Disease Effects of Evodia rutaecarpa and In Silico Discovery of Novel Lead Molecules" identifies the potential natural products isolated from Evodia rutaecarpa describes the possible mechanisms. 

In my opinion, the manuscript has multiple points which need to be addressed.

1. The title of the manuscript is misleading. The manuscript is completely on computational studies but the title indicates both experimental and computational studies.

2. The authors need to compare the obtained data with a few reference drugs.

3. Investigation of possible toxicity is missing.

4. Structure-activity relationship in the discussion part is missing.

5. The conclusion part is underdeveloped. The authors should include future directions. 

Round 2

Reviewer 1 Report

The authors have responded to all concerns meticulously and improved the manuscript accordingly. The revised draft is improved significantly. I do not have further comments. I recommend the revised draft for publication.

Reviewer 3 Report

The authors now considered all the comments and suggestions. I believe the manuscript is now suitable for publication.